# Effect of a family-involvement combined aerobic and resistance exercise protocol on cancer-related fatigue in patients with breast cancer during postoperative chemotherapy: study protocol for a quasi-randomised controlled trial

Chuhan Huang,[1] Yingjie Cai,[1] Yufei Guo,[1] Jingjing Jia,[2] Tieying Shi [iD] [1]

¹Department of Nursing, First Affiliated Hospital of Dalian Medical University, Dalian, China
²Qiqihar Medical College, Qiqihar, Heilongjiang, China

**Correspondence to**
Professor Tieying Shi;
sty11177@163.com

## ABSTRACT

**Introduction** Cancer-related fatigue (CRF) is one of the most common and debilitating side effects experienced by patients with breast cancer (BC) during postoperative chemotherapy. Family-involvement combined aerobic and resistance exercise has been introduced as a promising non-pharmacological intervention for CRF symptom relief and improving patients' muscle strength, exercise completion, family intimacy and adaptability and quality of life. However, evidence for the practice of home participation in combined aerobic and resistance exercise for the management of CRF in patients with BC is lacking.
**Methods and analysis** We present a protocol for a quasi-randomised controlled trial involving an 8-week intervention. Seventy patients with BC will be recruited from a tertiary care centre in China. Participants from the first oncology department will be assigned to the family-involvement combined aerobic and resistance exercise group (n=28), while participants from the second oncology department will be assigned to the control group that will receive standard exercise guidance (n=28). The primary outcome will be the Piper Fatigue Scale-Revised (R-PFS) score. The secondary outcomes will include muscle strength, exercise completion, family intimacy and adaptability and quality of life, which will be evaluated by the stand-up and sit-down chair test, grip test, exercise completion rate, Family Adaptability and Cohesion Scale, Second Edition-Chinese Version (FACESII-CV) and Functional Assessment of Cancer Therapy -Breast (FACT-B) scale. Analysis of covariance will be applied for comparisons between groups, and paired t-tests will be used for comparison of data before and after exercise within a group.
**Ethics and dissemination** This study has been approved by the Ethics Committee of the First Affiliated Hospital of Dalian Medical University (PJ-KS-KY-2021-288). The results of this study will be published via peer-reviewed publications and presentations at conferences.
**Trail registration number** ChiCTR2200055793.

## STRENGTHS AND LIMITATIONS OF THIS STUDY

⇒ This will be the first clinical study to explore the pre-liminary effects of a family-involvement combined aerobic and resistance exercise protocol on cancer-related fatigue in patients with breast cancer during postoperative chemotherapy.
⇒ The family members of the intervention group will record activities in a supervision diary and check in a WeChat group throughout the intervention to illustrate the role of the family members in the intervention process.
⇒ The main limitation of this study will be the small sample size.
⇒ The findings of this study may only reflect the characteristics of patients in this location given the limited study sites.
⇒ Another potential limitation is that study protocol lacks a long-term follow-up to assess the ongoing effects of family-involvement exercise.

## INTRODUCTION

According to the latest data published for 2020, the number of newly diagnosed cases of breast cancer (BC) in that year reached 2.26 million worldwide and the number of deaths due to BC was 680 000. BC has officially replaced lung cancer as the most common cancer worldwide and ranks first among females as the cause of cancer-related deaths.[1] In China, approximately 368 000 new BC cases were diagnosed in 2018, and approximately 416 000 new cases of BC were diagnosed in 2020,[2] showing an increasing trend year by year.[3] At present, patients with BC in China tend to be younger than those in other populations.[4] Overall, the number of BC survivors has increased, as the 5-year survival

rate of patients with BC has reached 90%. This has been attributed to the progress made in screening, diagnosis and therapeutic strategies applied in BC management.[5]

As a common adjuvant therapy after BC surgery, chemotherapy can improve the survival rate and prolong the survival time of patients with BC.[6 7] However, patients with low immunity during postoperative chemotherapy are prone to various discomforts, which increase their physical and mental burden and reduce their quality of life. CRF is one of the most common symptoms experienced by patients with BC during postoperative chemotherapy.[8] According to previous research,[9] 80%–90% of patients with cancer will experience CRF during treatment, and this condition will persist for a long time. Research has also shown that CRF not only increases the risk of accidental falls in patients with BC but also reduces their tolerance to chemotherapy drugs and increases the incidence of adverse reactions such as nausea and vomiting.[10] In addition, compared with other symptoms such as pain and depression caused by cancer, CRF has a major negative impact on patients' social interactions and re-employment.[11] Studies have shown that CRF in patients is significantly related to muscle strength[12] and health-related quality of life.[13] Therefore, it is necessary to improve CRF in patients with BC to promote their return to society and improve their quality of life.

Increased physical activity in patients with BC has been shown to improve CRF.[14] Physical activity refers to any physical movement that results in energy expenditure caused by skeletal muscle contraction,[15] including exercise training such as aerobic exercise, resistance exercise, balance exercise and flexibility exercise. Adult patients with BC aged 18–65 years are recommended to engage in regular physical activity at the intensity and for the duration specified by current guidelines.[16] Aerobic exercise and resistance exercise, as the most common exercise training methods in physical activity, have been proven to be safe and effective.[17] Studies have found that aerobic exercise combined with resistance exercise has the best effect on improving CRF.[18]

Although the benefits of combined aerobic and resistance exercise for CRF have been proven, many patients with BC do not exercise well. Good external support is one of the important factors for patients' adherence to exercise.[19] As an important part of external support for patients, family members are the main facilitators of daily activities and disease care during hospitalisation and after discharge.[20] Studies have shown that, compared with unsupervised exercise training, supervised exercise training with the participation of family members can better ensure the continuous and regular development of exercise, thereby maintaining and improving exercise effects.[21 22] In addition, increasing disease-related communication between family members and patients can effectively increase patients' confidence in treatment, improve the intimacy and adaptability between patients and their family members, and thereby, improve patients' family functioning.[23] Therefore, there is an urgent need for a method to promote the active support role of family members in the process of exercise training for patients with BC, which will then enhance the benefits of exercise in improving CRF in these patients.

The current study, therefore, proposes to assess the preliminary effects of a family-involvement exercise protocol on alleviating CRF in patients with BC through a quasi-randomised controlled trial (Q-RCT).

## METHODS AND MATERIALS

### Participant recruitment and eligibility criteria

Patients receiving postoperative chemotherapy for BC in a Grade A tertiary hospital in Dalian will be selected for this study.

The inclusion criteria will be: (1) pathological diagnosis of BC; (2) receiving first course of postoperative chemotherapy for BC; (3) age 18–65 years; (4) able to cooperate actively; (5) voluntary provision of informed consent to participate in this study; and (6) residing with family members during the intervention for 2 months. The exclusion criteria will be: (1) any other serious disease, such as cardiovascular disease, other type of cancer, and so on; (2) exercise contraindications, such as asthma, severe anaemia, disc herniation or other diseases; and (3) mental illness, previous mental illness or family history of mental illness.

One family member will be included for each patient with BC, and the inclusion criteria for family members will be: (1) age >18 years, with primary school education or above, and good communication skills; (2) spouse, parent or child of the patient with BC (immediate family member); (3) main caregiver as determined by the patient with BC and family members for at least 2 months; and (4) voluntary participation in the study. The exclusion criteria will be: (1) payment of remuneration for care of the patient with BC; and (2) a previous history of mental illness.

### Sample size

According to a previous study,[24] the score for CRF is expected to decrease by 2.29 points. According to the two-sided sample size test formula, a two-sided test with alpha=0.05 and 90% power will require a total sample of 23 patients in each of the two groups. To account for a 20% attrition rate, the required sample size is 56.

### Study design

To avoid contamination among the research participants, this study will adopt a lottery method to randomly divide the hospital's first and second oncology departments into a control group and intervention group and then select 28 patients from each ward who meet the requirements after screening by the inclusion and exclusion criteria.

The intervention team will consist of a sports rehabilitator, a senior clinical nursing specialist, a head nurse, three responsible nurses and a nursing graduate student. The sports rehabilitator will participate in the

formulation of the exercise programme. The nursing specialist will participate in the guidance of the experimental programme. The head nurse and three nurses will be responsible for quality control, identifying problems in the implementation of the programme, and making rectifications. The graduate student will be responsible for explaining the intervention to the patients with BC and their families as well as data collection and collation. Group members will jointly develop an exercise instruction manual (including exercise form, methods, precautions, etc) and record an instructional video for the exercise protocol.

The researcher will explain the purpose and implementation process of the study to the patients with BC when they are admitted to the hospital for their first chemotherapy treatment (T0). After their agreement to participate, participants will be asked to provide written informed consent. The participants will be informed that they can withdraw from the study at any time without any consequences. After obtaining informed consent from the patients, the researcher will conduct the stand-up and sit-down chair test and grip test with patients and issue general information questionnaires along with the R-PFS, FACESII-CV, and FACT-B. The intervention study will begin after the patients (and their family members) have completed their first assessment.

According to previous relevant studies,[25 26] exercise training begins to play a role in improving CRF during the 8-week exercise intervention process. Therefore, the patients will be tested with the stand-up and sit-down chair test and the grip test at the time of the fourth (T1) and eighth (T2) weekends of the intervention. The exercise completion rates in the two groups will be evaluated at these time points, and the researcher will again issue the general information questionnaires, R-PFS, FACESII-CV and FACT-B to evaluate the effects of the exercise intervention. In the process of distributing the questionnaires, if patients are unable to understand the questionnaire items, the researcher will explain the items. A Consolidated Standards of Reporting Trials flowchart for the study is presented in figure 1. The schedule for trial enrolment, intervention and assessment is presented in table 1.

## Control group

When patients are admitted to the hospital for their first chemotherapy treatment, the researcher will provide exercise guidance for 'combined aerobic and resistance exercise' and guide patients through the manual and video for exercise training, informing them of the recommended exercise time, form, method, intensity and precautions. The patients will be followed up monthly to learn their exercise status, and if patients have not persisted in following the recommendations, the researcher will ask patients why in order to help solve their problems and to encourage them to persist in exercise training.

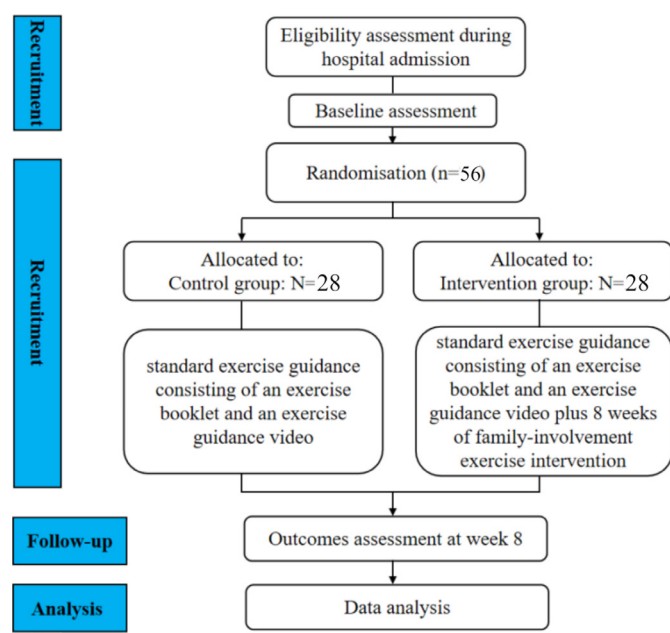

**Figure 1** A Consolidated Standards of Reporting Trials flow chart of the study.

## Intervention group

The family-involvement combined aerobic and resistance protocol to be applied in the intervention group is described in detail in table 2.

**Table 1** Schedule of trial enrolment, intervention and assessment

| Study period | | | |
|---|---|---|---|
| | Beginning of intervention (week 0) | Intervention period (week 4) | End of intervention (week 8) |
| Inclusion/ exclusion criteria | × | | |
| Informed consent | × | | |
| Demographic characteristics | × | | |
| Randomisation and allocation | × | | |
| R-PFS | × | × | × |
| Stand-up and sit-down chair test | × | × | × |
| Grip test | × | × | × |
| Exercise completion rate | | × | × |
| FACESII-CV | × | × | × |
| FACT-B | × | × | × |

FACESII-CV, Family Adaptability and Cohesion Scale, Second Edition-Chinese Version; FACT-B, Functional Assessment of Cancer Therapy-Breast; R-PFS, Piper Fatigue Scale-Revised.

**Table 2** Exercise training intervention with family members

| | | |
|---|---|---|
| Guidance for family members | Evaluate the patients' life and social environment through family members and explain to family members the importance of family members accompanying the patient during exercise. | |
| | Distribute exercise manuals to family members and play exercise instructional videos. | |
| | Instruct family members to master the exercise time, method, standard posture and precautions, as well as precautions for the use of elastic bands and dumbbells for patients with BC. | |
| Supervision of family members | Issue supervision diaries to family members* | Instruct and remind family members regarding the completion of the supervision diary through WeChat or phone every month. |
| | | On readmission to the hospital, the patient will be required to bring the supervision diary in order for the record in the diary to be checked. |
| | Supervision of family members via WeChat | Establish a special WeChat group for family members, send exercise-related health knowledge to the WeChat group every week and regularly remind family members to urge patients to exercise. |
| | | Encourage family members to ask questions actively in the WeChat group and provide timely feedback on the supervision situation and their feelings, answering relevant questions from family members in the process of guiding and supervising patients. |
| | | Provide appropriate praise and rewards to the family members who complete the monitoring goals every week first, forming a competition mechanism to stimulate interest. |
| Follow-up with family members | Make phone calls or complete face-to-face follow-up with family member monthly, asking them about questions they encountered during the care of the patients and answering those questions. | |
| | Ask the patient about the training situation. If patients' exercise completion are poor, further inquire about why and analyse the reasons, helping the family to solve the problem. | |
| Instructions to family members | | |
| Family members will accompany patients | Participate in the whole process of the patient's exercise protocol and provide the patient with full emotional support. Before exercising, choose the method of aerobic exercise together with the patient according to the patient's preference and play music to help the patient relax while they exercise. | |
| Family members will supervise patients | Give exercise guidance to the patient, informing the patient of the correct exercise method by playing the provided video. When the patient makes an error or non-standard movement during the exercise, provide timely correction. | |
| | Supervise the patient exercising. Take photos or videos while the patient is moving to record the patient's movement. Keep a supervision diary after each exercise session and upload exercise photos or videos in) the WeChat group to ensure the effect of family supervision. | |
| Family members will encourage patients | Encourage and support the patient to enhance their confidence; encourage the patient to describe difficulties encountered during exercise and ensure them that the patient and family member will overcome the difficulty together. | |

*The first part of the supervision diary includes basic information, including initials of family members, initials of patients, the relationship between family members and patients, patients' height and weight, initials of the doctor in charge, date of chemotherapy, and so on. This part is filled out by the researcher before the diary is distributed to the family members. The second part includes the exercise date, time, location, exercise duration, frequency and number of groups, the patient's exercise status and whether the family member participated in the exercise guidance and supervision throughout the exercise.
BC, breast cancer.

## Combined aerobic and resistance exercise

The designed exercise training intervention will last for 8 weeks, and each patient can select the most comfortable period of the week according to his or her own situation every week. Each training time is scheduled for half an hour after a meal, and participants are instructed

to avoid training with an empty or full stomach. Each workout includes aerobic and resistance exercises that are both progressive. A rest period of 5 min is allowed between aerobic and resistance exercise. Aerobic exercise is accomplished by cycling, walking on a treadmill or brisk walking on the ground, depending on the patient's preference, according to the following schedule: twice per week for 15–20 min in weeks 1–4 and three times per week for 25–35 min in weeks 5–8. For resistance exercise, the patients will use 0.45 kg dumbbells or a mineral water bottle filled with 500 mL of water and a 18 cm brown elastic band with the brand name TheraBand. These items will be provided by the researchers as the main exercise tools for four resistance exercise movements: hip abduction, seated knee raising, raise before standing and standing lateral raise. These exercises are to be completed twice per week in weeks 1–4, with two sets of each exercise and each set consisting of 8–12 repetitions and then three times per week in weeks 5–8, with three sets of each exercise (each set includes 8–12 repetitions), with plenty of rest between sets.[27–30]

### Hip abduction
This exercise is done in a standing position, with the elastic band around the ankle of one foot while the other foot is on the elastic band. The opposite hand holds the seat, and the supporting foot is firmly planted on the ground to keep the trunk stable. Initially, the buttocks exert force as the elastic band is stretched outward to the side by raising the leg until the angle between the practice leg and the supporting foot is about 30°. After a pause for about 1 s, the leg is slowly lowered to the ground. The exercise is done by alternating the left and right feet between sets.

### Seated knee raising
This exercise is done in a sitting position with a straight back while holding the seat with both hands, keeping the upper torso upright, stepping on one end of the elastic band with one foot and pulling the elastic band around the ankle of the other foot. The knee is then lifted to the maximum angle of joint activity and the after a pause for about 1 s, slowly lowered. The exercise is done by alternating the left and right legs between sets.

### Raise before standing
While standing with feet parallel, head straight and looking straight ahead, the dumbbells are held at both sides of the body by both hands. Both arms are then raised in front of the body, and after a pause for about 1 s, the arms are slowly lowered. Participants are instructed to inhale during the horizontal lift and exhale while returning the arms to their original positions.

### Standing lateral raise
While standing with feet parallel, head straight and looking straight ahead, the dumbbells are held at both sides of the body by both hands. The arms are raised from the sides, with elbows slightly bent and focus placed on the shoulders. Once the arm is raised to a horizontal position, participants pause for about 1 s before slowly lowering the arms to the initial positions. Participants are instructed to inhale during the horizontal lift and exhale while returning the arms to their original positions.

The intensity of aerobic exercise should be moderate; that is, the Borg subjective fatigue scale score should reach 13–14 points (feeling a little hard). The intensity of resistance exercise should be tolerable to the patient. If obvious fatigue and muscle pain occur during or after training, the intensity of physical activity should be appropriately reduced. If patients have severe pain, discomfort and other symptoms, they should immediately stop the activity and be admitted to the hospital for re-examination.[31 32]

During the exercise, family members are instructed to pay attention to the patient's reactions and to adjust the elastic band or exercise speed and intensity as needed. They must also ensure the patient is not straining or using too much force during exercise to prevent stretching the wound or the place where the catheter is placed. They should also pay attention to the patient's movement at all times. If the patient experiences dizziness, discomfort or other adverse events, they should be promptly instructed to stop exercising and visit the hospital if necessary.

### Outcome measurements
The outcome measurements for this pilot study will be derived from baseline assessments and clinical outcomes.

### Demographic and clinical characteristics of the participants
A self-designed demographic and clinical data form will be employed to collect the participants' sociodemographic data (eg, age, educational background, employment status, marital status and household income), the participants' medical history (eg, date of BC diagnosis, current stage of BC and date and type of treatment) and family members' sociodemographic data at baseline (T0).

### Primary outcome: CRF
#### Piper Fatigue Scale-Revised (R-PFS)[33]
The R-PFS will be applied to assess the participants' subjective fatigue. This self-administered questionnaire contains 22 items with scores ranging from 0 to 10 and includes four domains of subjective fatigue. The R-PFS is available in a simplified Chinese version,[34] with high reliability and validity, and has high reliability in the population with BC.

### Secondary outcomes: muscle power, exercise completion, family intimacy and adaptability and quality of life
Muscle strength, exercise completion, family intimacy and adaptability and patient quality of life will be assessed as secondary outcomes at T0, T1 and T2 using the stand-up and sit-down chair test, grip test, exercise completion rate, FACESII-CV and FACT-B.

### Stand-up and sit-down chair test (number of times standing up from the chair within 30 s)[35]

This test will be used to evaluate the leg strength of the participant. The test procedure is as follows: (1) an upright chair (or a folding chair) is placed against a wall (for the sake of safety); (2) the participant sit in the middle of the chair with the right and left feet shoulder-width apart, and one foot may be put slightly forward and the other slightly back, while both arms are crossed at the waist and held near the chest; (3) during the test, participants should completely stand up and then completely sit down; (4) the number of times that the participant stands up from and sits down in the chair within 30 s is recorded; and (5) for safety purposes or when necessary, the participant may use his or her arms for assistance.

### Grip test[32]

This test will be used to evaluate the arm strength of the participants. A CAMRY (model EH101) electronic grip strength metre is used in units of kg/lb. The maximum force is 90 kg/198 lb, and the division value is 0.1 kg/0.2 lb. Participants are in a standing position, with their hands hanging down naturally, while their feet are positioned under the shoulders and hands are held away from the body.

### Exercise completion rate

This test will be used to assess the performance of the two types of exercise. The exercise completion rate in this study will be calculated by dividing the number of participants in the intervention group who completed 18 or more training sessions (more than 90% of the total training volume) by the total number of people in the intervention group and multiplying that value by 100%.

### FACESII-CV[36]

The FACESII-CV will be adopted to assess the participants' family intimacy and adaptability. A higher score indicates higher intimacy and adaptability. The FACESII-CV is available in a simplified Chinese version, with high reliability and validity.

### FACT-B[37]

The FACT-B will be adopted to assess the patients' quality of life. A higher score reflects better quality of life. The FACT-B is available in a simplified Chinese version, with adequate psychometric properties reported among patients with BC.

### Data analysis

Statistical analysis of the results will be completed using SPSS software V.25.0. Analysis of covariance will be applied for comparisons between groups, and paired t-tests will be used for comparisons from before to after exercise within a group. Values of $p < 0.05$ will represent that a significant difference has been detected between test results.

### Data management

After data collection is complete, all handwritten data will be converted to electronic data. All data will be independently recorded by two researchers in Excel spreadsheets. The software will automatically check for inconsistent or problematic data based on the inspection results and generate a data problem table. After all data have been confirmed, reconciled and stored in an electronic database, participants' identifying information (eg, real name) will not appear in the relevant reports of the trial to protect their privacy. Only researchers directly involved in the analysis of this study will have access to the final trial dataset, which will contain only coded data.

### Data sharing statement

A technical appendix, statistical codes and dataset will be available at any time on request to the corresponding author.

### Patient and public involvement

Patients or the public were not involved in the design of this study and will not be involved in conducting the research or reporting the results.

### Ethics and dissemination

Ethical approval of the proposed study has been granted by the Ethics Committee of the First Affiliated Hospital of Dalian Medical University (PJ-KS-KY-2021-288), and the trial has been registered in the Chinese Clinical Trial Registry (registration number: ChiCTR2200055793). Only participants who provided written informed consent will be included in the study. The results of this study will be published via peer-reviewed publications and presentations at conferences.

## DISCUSSION

As one of the most common symptom clusters in patients with BC, CRF can significantly diminish patients' quality of life and daily functioning.[38] An increasing number of studies has demonstrated that combined aerobic and resistance exercise has beneficial effects on symptom management in patients with cancer.[39] However, due to differences in the time and intensity of exercise as well as individual differences, the effect of such exercise has varied among different studies. Gokal et al[40] tested a home-based walking intervention based on self-management in 25 patients with BC receiving chemotherapy and found that this self-management exercise intervention could improve patients' fatigue symptoms as well as their physical activity level. However, some patients had difficulty adhering to exercise training, causing them to stop exercising. Uhm et al[41] conducted a 12-week RCT on 356 patients with BC, which showed that combined aerobic and resistance exercise improved patients' fatigue symptoms, muscle strength and quality of life. However, because some patients could not persist in completing the exercise, the advantages of the tested exercise over traditional exercise programmes were not obvious. Previous studies have confirmed that family-involvement exercise interventions have many benefits

for patients after stroke,[42] coronary heart disease,[43] lung cancer[44] and other conditions, but there are few reports on the effect of family-involvement exercise interventions in patients with BC in China. This highlights a great need to explore the effects of family-involvement exercise on CRF in patients with BC.

The principal strength (and novel aspect) of this study resides in the fact that in the intervention, each participant will be able to choose the type of aerobic exercise they prefer. Also, the study will provide exercise instruction videos designed to increase patients' motivation to exercise. In addition, we aim to provide new evidence on the cost-effectiveness of this type of intervention (to our knowledge, virtually no evidence has previously been reported in this regard). Moreover, an exercise booklet will be provided to the participants in both the intervention and control groups. The information presented in this booklet has been comprehensively adapted from relevant national guidelines, professional bodies and research evidence in published peer-reviewed articles.

The proposed study will also have some limitations. Given the limited number of study sites, the study sample may not offer a widely representative sample of patients with BC. Moreover, due to the visible nature of the exercise intervention, blinding of the participants and investigators cannot be performed in this study, which might increase the risk of detection bias during the study's implementation. The lack of long-term follow-up to assess the ongoing effects of family-involvement exercise might be another limitation, but this can be considered in the future full-scale trial as one of the main study outcomes. Furthermore, this study will not be a rigorous RCT but rather a Q-RCT, and a future multicentre, large-scale RCT will need to be conducted to further verify the research evidence for the effects of family-involvement exercise on the management of patients with BC with CRF.

The proposed Q-RCT will assess the preliminary effects of a family-involvement exercise programme for alleviating CRF in patients with BC undergoing postoperative chemotherapy. The convenience of the family-involvement exercise for the management of CRF may provide patients, healthcare professionals and policymakers with further guidance for CRF management in the long-term.

**Contributors** CH conceived and designed the study, and TS will oversee the research team and research process. YC and CH will be the main implementers of the study and drafted this manuscript. JJ wrote the ethics review confirmation. YG participated in the design of the study and assisted in drafting this manuscript. All authors have read and agreed to the final version of the manuscript.

**Funding** Preparation of this study protocol was funded by the Dalian Science and Technology Innovation Fund Science and Technology Benefit People Project (Award/ Grant No. 2022JJ13FG109). The authors have not declared a specific grant that will fund the proposed research from any funding agency in the public, commercial or not-for-profit sectors.

**Competing interests** None declared.

**Patient and public involvement** Patients and/or the public were not involved in the design, or conduct, or reporting or dissemination plans of this research.

**Patient consent for publication** Consent obtained directly from patient(s).

**Provenance and peer review** Not commissioned; externally peer reviewed.

**ORCID iD**
Tieying Shi http://orcid.org/0000-0002-8599-3587

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
