## [Reviewer comments · BMJ Open]

ARTICLE DETAILS

TITLE (PROVISIONAL)	Effect of a family-involvement combined aerobic and resistance exercise protocol on cancer-related fatigue in breast cancer patients during postoperative chemotherapy: study protocol for a quasi-randomized controlled trial
AUTHORS	Huang, Chuhan; Cai, Yingjie; Guo, Yufei; Jia, Jingjing; Shi, Tiejing

VERSION 1 – REVIEW

REVIEWER	Han, Lin Lanzhou University
REVIEW RETURNED	24-Aug-2022

GENERAL COMMENTS	Thank you for the opportunity to review this study protocol. Cancer-related fatigue (CRF) is one of the most common and debilitating side effects in patients with breast cancer (BC) throughout postoperative chemotherapy. And some studies have already confirmed that aerobic and resistance exercise can be beneficial to CRF in breast cancer patients. The topic is really important and the methodological framework used is very robust and well described. However, some non-secondary aspects need to be clarified in my opinion. 1. On page 3 it is not clear why patients over 65 years old are excluded. Conducting a family-involvement aerobic combined resistance exercise this is not necessary and why choose to exclude the elderly?2. On page 4 the patients' test will conduct at the time of the second admission chemotherapy and the fourth admission chemotherapy, why? Give a clear explanation.3. The difference between the exercise style of the intervention group and the control group is too large, so how to show that the exercise style of the intervention group is effective because of the family involvement rather than the role of aerobic combined resistance exercise itself?
---

REVIEWER	Gupta, Ananya National University of Ireland, Physiology
REVIEW RETURNED	20-Sep-2022

GENERAL COMMENTS	The current protocol does not include an outcome measure to determine aerobic capacity. A 6 minute walk test is an easy test for assessment of aerobic capacity. A test for aerobic capacity and VO ₂ must be included in the protocol. Poor aerobic capacity
--

	contributes to fatigue and it is extremely important to assess this at the beginning and end and also for monitoring progress through the study.
--	--

VERSION 1 – AUTHOR RESPONSE

Part A (Reviewer 1: Dr. Lin Han).

The reviewer’s comment:

Dr. Lin Han, Lanzhou University

Comments to the Author:

Thank you for the opportunity to review this study protocol. Cancer-related fatigue (CRF) is one of the most common and debilitating side effects in patients with breast cancer (BC) throughout postoperative chemotherapy. And some studies have already confirmed that aerobic and resistance exercise can be beneficial to CRF in breast cancer patients. The topic is really important and the methodological framework used is very robust and well described. However, some non-secondary aspects need to be clarified in my opinion.

1. On page 3 it is not clear why patients over 65 years old are excluded. Conducting a family-involvement aerobic combined resistance exercise this is not necessary and why choose to exclude the elderly?
2. On page 4 the patients’ test will conduct at the time of the second admission chemotherapy and the fourth admission chemotherapy, why? Give a clear explanation.
3. The difference between the exercise style of the intervention group and the control group is too large, so how to show that the exercise style of the intervention group is effective because of the family involvement rather than the role of aerobic combined resistance exercise itself?

The authors’ Answer:

Thank you very much for your constructive comments and suggestions on our manuscript!

1. On page 3 it is not clear why patients over 65 years old are excluded. Conducting a family-involvement aerobic combined resistance exercise this is not necessary and why choose to exclude the elderly?

Reply:

Thank you very much for this important comment.

At present, breast cancer patients in China tend to be younger. According to the literature, 8% to 10% of breast cancer patients are diagnosed at the age of less than 35 years old. In addition to the huge population base, the number of new patients in this age group can reach 30,000 to 40,000 each year, and the number of patients younger than 40 years old is close to 50,000. In addition, the recommended levels of physical activity in the guidelines on “Lifestyles of Breast Cancer Patients” apply to patients aged 18-64. Although patients over the age of 65 should exercise at the recommended level of physical activity, this population is more likely to have chronic medical conditions that limit mobility. Therefore, patients over 65 years of age are excluded from this study protocol.

The content “And at present, breast cancer patients in China tend to be younger.” and “Adult breast cancer patients aged 18-65 should engage in regular physical activity at the time and intensity recommended by the guidelines.” have been modified. (Please see the revised manuscript at line19, page 3 and line4-6, page 4).

The reference “[4]Jiang Xiaoting, Liu Qiang, Song Erwei. Questions and reflections in the diagnosis and treatment of younger breast cancer in China[J]. National Medical Journal of China, 2022, 102(36):

2823-2827.” and “[16]Breast Health Group(BEST: Breast Education Screening Diagnosis and Treatment Group) of the Branch of Women Health of Chinese Preventive Medicine Association. Guidelines on life-style modification for Chinese breast cancer survivors[J]. Chinese Journal of Surgery, 2017, 55(2): 81-85.” have been added.

2. On page 4 the patients' test will conduct at the time of the second admission chemotherapy and the fourth admission chemotherapy, why? Give a clear explanation.

Reply:

Thank you very much for this important comment.

According to previous relevant studies, exercise training begins to play a role in “improving cancer-related fatigue” during the 8-week exercise intervention process. We modified the testing time of patients according to previous related studies: In this study, patients were tested on the fourth and eighth weekends of the intervention to observe the effect of exercise training during and after the intervention.

The content “According to previous relevant studies, exercise training begins to play a role in “improving cancer-related fatigue” during the 8-week exercise intervention process. Therefore, the patients are tested with the stand-up and sit-down chair test and the grip test at the time of the fourth (T1) and eighth (T2) weekends of the intervention. ” has been modified. (Please see the revised manuscript at line36-37, page 5 and line1-2, page 6).

The reference “[25]Jiang Yongqin, Yan Ling, Liu Chunyan, et al. Effects of exercise prescription on cancer-related fatigue in breast cancer patients[J]. Chinese Journal of Nursing, 2008(10): 906-909.” and “[26]Yan Ling. Effects of exercise prescription on cancer-related fatigue in breast cancer patients[D].Tianjin Medical University, 2008.” have been added.

3. The difference between the exercise style of the intervention group and the control group is too large, so how to show that the exercise style of the intervention group is effective because of the family involvement rather than the role of aerobic combined resistance exercise itself?

Reply:

Thank you very much for this important comment.

The specific exercise methods of “aerobic combined resistance exercise” performed by the intervention group and the control group are the same, and the difference between the two groups is only in the presence or absence of family members participating in the exercise.

We have described this part more clearly in the revised manuscript.

The content “When patients are admitted to the hospital for the first chemotherapy, the researcher provides exercise guidance for them with ‘aerobic combined resistance exercise’, and guides patients exercise training through the manual and video, and informs exercise time, form, method, intensity as well as precautions.” and “Based on the control group, ‘aerobic combined resistance exercise’ with family members is presented in Table 2” have been modified. (Please see the revised manuscript at line15,16,22, page 6).

We appreciate for your warm work earnestly, and hope that the correction will meet with approval. Once again, thank you very much for your comments and suggestions!

Part B (Reviewer 2: Dr. Ananya Gupta).

The reviewer's comment:

Dr. Ananya Gupta, National University of Ireland

Comments to the Author:

The current protocol does not include an outcome measure to determine aerobic capacity. A 6 minute walk test is an easy test for assessment of aerobic capacity. A test for aerobic capacity and VO₂ must be included in the protocol. Poor aerobic capacity contributes to fatigue and it is extremely important to assess this at the beginning and end and also for monitoring progress through the study.

The authors' Answer:

Thank you very much for your constructive comments and suggestions on our manuscript!

As you said, the 6 minute walk test and peak VO₂ can assess the patients' aerobic capacity, and after searching the relevant literature, it is found that assessing the patients' fatigue level through the aerobic capacity level is crucial. However, due to the limitations of the research equipments and environment, this study does not have objective conditions to measure the aerobic capacity of patients, so these tests are not included in the protocol. We will revise the protocol based on your comments and conduct further research if conditions permit in future research.

We appreciate for your warm work earnestly, and hope that the protocol will meet with approval. Once again, thank you very much for your comments and suggestions!